# Tramadol vs. Lidocaine Administered Intraperitoneally and in Incisional Lines for the Intraoperative and Postoperative Pain Management of Romifidine-Telazol-Anesthetized Swine Undergoing Umbilical Hernia Repair

**DOI:** 10.3390/ani13182905

**Published:** 2023-09-13

**Authors:** Giovanna Lucrezia Costa, Filippo Spadola, Simona Di Pietro, Vincenzo Nava, Patrizia Licata, Elisabetta Giudice, Fabio Leonardi, Fabio Bruno, Laura Messina, Francesco Macrì, Daniele Macrì, Vincenzo Ferrantelli, Marco Tabbì, Claudia Interlandi

**Affiliations:** 1Department of Veterinary Sciences, University of Messina, Via Palatucci Annunziata, 98168 Messina, Italy; fspadola@unime.it (F.S.); simona.dipietro@unime.it (S.D.P.); vnava@unime.it (V.N.); plicata@unime.it (P.L.); egiudice@unime.it (E.G.); fabio.bruno@studenti.unime.it (F.B.); laura.messina@studenti.unime.it (L.M.); fmacri@unime.it (F.M.); marco.tabbi@unime.it (M.T.); cinterlandi@unime.it (C.I.); 2Department of Veterinary Science, University of Parma, Via del Taglio 10, 43126 Parma, Italy; fabio.leonardi@unipr.it; 3Zooprophylactic Institute, Via Gino Marinuzzi 4, 90100 Palermo, Italy; daniele.macri@izssicilia.it (D.M.); vincenzo.ferrantelli@izssicilia.it (V.F.)

**Keywords:** anesthesia, hernia, local, lidocaine, pain, swine, tramadol

## Abstract

**Simple Summary:**

Umbilical hernia in swine is a congenital condition that may require surgical treatment. Locoregional anesthesia is commonly used in livestock to provide analgesia for surgery. However, the relatively low pH of inflamed tissue may reduce, slow down, or compromise the efficacy of local anesthetics. Phlegmon, ulcers, and dermatitis are very common lesions in livestock, that result in an inflammatory process with pH reduction of the tissues. Consequently, a locally effective opioid may replace or compliment the analgesic efficacy of local anesthetics. Tramadol is a weak opioid with good analgesic efficacy and few side effects. The results of the study showed that tramadol could be used for pain management in livestock.

**Abstract:**

The aim of the study was to compare the analgesic efficacy of tramadol and lidocaine for local anesthesia during umbilical hernia repair in swine. The study was performed on 66 large white crossbred swine. The swine received a mixture of tiletamine/zolazepam at 5 mg/kg and romifidine at 80 µg/kg, administered intramuscularly. Then, they were divided into three groups (*n* = 22) that received different treatments with lidocaine at 4 mg/kg and tramadol at 4 mg/kg. The LL group received lidocaine both by infiltration of the surgical planes and intraperitoneally. The LT group received lidocaine by infiltration of the surgical planes and tramadol intraperitoneally. The TT group received tramadol both by infiltration of the surgical planes and intraperitoneally. In all groups, the infiltration of the surgical planes into the umbilical region involved both the skin and muscle planes. Heart rate, noninvasive arterial blood pressure, and respiratory frequency were recorded during surgery. The response to the surgical stimulus was evaluated using a cumulative pain scale (the cut-off point for rescue analgesia was set to 10). Postoperative pain was assessed using the UNESP-Botucatu pig composite acute pain scale (the cut-off point was set to 4). The trend of physiological variables was adequate for patients under anesthesia. No subject required intraoperative and postoperative rescue analgesia. Tramadol could therefore be used for pain management in livestock.

## 1. Introduction

Umbilical hernia in swine is a congenital paratopia with an incidence ranging from 0.13% to 5% [1,2,3]. The displacement of the abdominal contents near the umbilical region through the hernial ring results in the development of a protuberance in which wounds, ulcers, and infections develop, compromising the welfare and economic aspects of breeding [4,5,6].

The use of local anesthetics and opioids administered on-site for intraoperative and postoperative pain management has been used in human medicine for many years and is also becoming increasingly popular in veterinary medicine [7,8,9,10]. In new multimodal analgesia techniques, local anesthetics, alone or in combination with opioids, play a crucial role in incisional blocks and intraperitoneal administration [11,12].

Locoregional anesthesia is the main anesthetic technique used in livestock since most surgeries are performed with the animals standing. However, umbilical hernia surgery requires general anesthesia or deep sedation, as the patient’s dorsal recumbency is required [13]. Deep sedation combined with locoregional anesthesia is the most used anesthetic technique for this purpose to manage the intraoperative and postoperative pain [10,14,15,16].

Lidocaine is the main local anesthetic used in livestock due to its low cost [13]. The analgesic and anesthetic effects of lidocaine are due to its ability to block sodium/calcium channels [10,17,18]. However, the efficacy of local anesthetics can be reduced, slowed down, or nullified by a relatively low pH [18]. Lidocaine is a poorly water-soluble weak cationic base. After intravenous administration, it is largely bound to albumin and, together with other plasma transport proteins, has a distribution volume of approximately 91 L/kg.

The low pH of the lung, but especially the action of cytochrome P450 oxidase (CYP450), seems to contribute to the elimination of the molecule from the bloodstream. Lidocaine is mainly metabolized in the liver by CYP450, which produces active metabolites such as monoethylglycylxylidide, *N*-ethylglycine, and glycylxylidide. The accumulation of these metabolites block the lidocaine metabolism pathway, leading to possible cases of intoxication [19,20]. These metabolites are eliminated by the kidneys at a rate of approximately 0.85 L/kg/h. The metabolic rate determines the half-life of lidocaine and consequently contributes to its accumulation [19,20].

Lidocaine blocks impulse propagation in nerve fibers by acting on voltage-gated sodium channels, reducing neurotransmitter release due to the incorrect activation of presynaptic calcium channels [21]. This mechanism of action results in a wide range of effects, mainly antithrombotic, but also antimicrobial and antitumor [22].

Lidocaine has anti-arrhythmic and anti-inflammatory properties [23]. Various studies have highlighted that lidocaine is able to limit tissue damage when the inflammatory response is initiated [23]. Its action inhibits macrophage and neutrophil activity by reducing the release of pro-inflammatory cytokines (leukotrienes, interleukins IL-1β, IL-4, IL-6, TNF-α, and histamine) through a reduction of ATP-induced increase in intracellular Ca^2+^, which activates p38 MAPK that reduces ion exchange through membrane channels [20]. The activation of leukocytes and glial cells together with signaling cytokines results in the activation of tissue nociceptors. The nociceptive agonist action of lidocaine results in its analgesic effect in acute and chronic pain [24]. Lidocaine administration was able to reduce tactile allodynia through a decrease in pro-inflammatory cytokines, according to in vivo studies in a rat model of neuropathic pain [25].

Lesions such as phlegmon, ulcers, and dermatitis are common in livestock and lead to an inflammatory process with a relatively low pH [26].Consequently, a locally active opioid can replace or compensate the analgesic efficacy of local anesthetics [8,9,14,27,28,29]. Tramadol is a weak µ-opioid receptor agonist and occurs as a racemic mixture of two enantiomers. The positive enantiomer inhibits serotonin reuptake, while the negative enantiomer inhibits norepinephrine reuptake. Both enantiomers of tramadol act as µ-opioid receptor agonists, resulting in a synergistic effect that produces effective analgesia. Tramadol has minimal side effects on the cardiovascular, respiratory, and gastrointestinal systems. Tramadol is eliminated unchanged mainly through feces and urine (99%), and only a very small amount (0.02%) is excreted in milk. Tramadol combined with an alpha2-adrenoceptor agonist has been previously used in large animals. This combination represents a viable option to achieve deep and balanced sedation [30,31,32,33,34,35].

The aim of this study was to compare the analgesic efficacy of tramadol and lidocaine administered by infiltration of the surgical planes and intraperitoneally during umbilical hernia repair in swine. The study’s hypothesis was that locally administered tramadol could be used as an alternative to lidocaine for managing surgical pain in livestock.

## 2. Materials and Methods

The study was approved by the ethics committee of the University of Messina (protocol N 027/2018) and was conducted in compliance with Italian law (DM 116192), European law (GU ECL 358/1 18/12/1986), and US laws (Animal Welfare Assurance No A5594-01, Department of Health and Human Services, Washington, DC, USA) according to Legislative Decree no. 193 of 6 April 2006. Pharmacological treatments were recorded in the appropriate barn register, and informed consent was obtained from the swine owners. No slaughter products were intended for trade.

### 2.1. Animals

The study was performed on 66 large white crossbred swine, aged 60 ± 2 days, and weighing 25 ± 10 kg, including 18 males and 48 females. The inclusion criterion was the presence of an umbilical hernia 3–5 cm in diameter. The swine were randomly divided, by draw, into three groups of 22 subjects each: LL, LT, and TT groups. The presence of omphalites was the exclusion criterion.

### 2.2. Treatment Administration

The swine were weighed with a scale (Zoopiro, Cutro, Italy) and placed in a pen for 30 min to let them acclimatize. Then, a combination of tiletamine/zolazepam at 5 mg/kg (Zoletil, Virbac, Carros, France) and romifidine at 80 µg/kg (Sedivet, Boeringher, Ingelheim am Rhein, Germany) was administered, in the same syringe, intramuscularly to induce anesthesia.

After the surgical anesthesia stage and muscle relaxation were achieved, swine were placed in dorsal recumbency, the umbilical region was aseptically prepared, and a local analgesic protocol consisting of 5% tramadol (Altadol, Formevet, Milan, Italy) and 2% lidocaine (Lidocaine, Esteve, Barcelona, Spain), both at 4 mg/kg, was performed.

The LL group received lidocaine both by infiltration of the surgical planes and intraperitoneally. The LT group received lidocaine by infiltration of the surgical planes and tramadol intraperitoneally. The TT group received tramadol both by surgical plane infiltration and intraperitoneally.

In the LL and TT groups, the overall dose of the drug was divided into two doses, one administered by infiltration of the surgical plane and the other administered intraperitoneally. To ensure homogeneous tissue infiltration, the volume of the drug, intended for each surgical plane (by infiltration or intraperitoneally), was increased to 40 mL by adding physiological solution. In all groups, the administration by infiltration of the surgical planes in the umbilical region involved both the skin and muscle planes, while the intraperitoneal injection was performed in the hernial sac.

During the surgery, the swine received physiological solution (0.9% sodium chloride) at 5 mL/kg/h using an infusion set with a flow regulator from 5 to 250 mL/h (Medvet, Srl, Taranto, Italy), through a venous catheter, 14 G × 5/12 (Medvet, Srl, Taranto, Italy), placed in the jugular vein.

### 2.3. Umbilical Hernia Repair

An elliptical skin incision was made and any adhesions between the parietal peritoneum and the skin were removed. Displaced organs were properly repositioned and the hernia ring was exposed and refreshed. The linea alba was closed with 2-0 chromic catgut (Chromic catgut absorbable monofilament suture, Catgut chrom^®^, B-Braun, Melsungen, Germany) with an interrupted horizontal mattress suture. An autologous flap was created from the remaining hernia sac. The subcutaneous tissue was sutured with 2-0 chromic catgut with simple interrupted stitches, and the excess skin was removed and sutured with 2-0 nylon (Nylon no absorbable monofilament suture, DACLON NYLON TR^®^, Vetefarma, Cuneo, Italy) with simple interrupted stitches. Herniorrhaphy was carried out by the same two experienced surgeons in collaboration [36,37,38].

### 2.4. Measurement of Physiological Parameters

During anesthesia, heart rate (HR), systolic (SAP), diastolic (DAP), and mean arterial blood pressure (MAP) (all non-invasive: applied a cuff of 12–19 cm size to arm), as well as hemoglobin oxygen saturation (SPO_2_%) (applied the sensor to tongue) and intrarectal body temperature (T°) were recorded using a CAMS 2 multiparameter monitor (Forlì, Italy). Respiratory frequency (*f_R_*) was assessed by chest excursions.

These parameters were recorded at T1 (baseline; after sternal recumbency), T5 (five minutes after dorsal recumbency), T10 (skin incision), T15 (muscular plane incision), T20 (hernial sac opening and herniorrhaphy), T25 (muscular and subcutaneous plane suture), and T30 (skin suture).

### 2.5. Assessment of Response to Surgical Stimulus

The intraoperative response to surgical stimulus was assessed using a cumulative pain scale (CPS). A numerical score ranging from 0 to 4 was assigned based on the percentage variation from baseline (T1) values of each measured physiological parameter (*f_R_*, HR, and SAP) at T10, T15, T20, T25, and T30, according to the following scheme: 0 = ≤0%, 1 = >0% but ≤10%, 2 = >10% but ≤20%, 3 = >20% but ≤30%, and 4 = >30%. The sum of the scores for the three parameters represented the intraoperative pain score. If the sum of the CPS scores exceeded 10, corresponding to a 20% increase in the three physiological parameters, a 2 mg/kg lidocaine was administered by infiltration of the surgical planes and intraperitoneally [16,33,39].

Postoperative pain was assessed using the UNESP-Botucatu pig composite acute pain scale (UPAPS) [36]. Scores were assigned by three observers unaware of the treatment received by the swine, from the recovery time in a standing position to six hours later (R0, R1, R2, R3, R4, R5, and R6). The cut-off point for administration of rescue analgesia, which consisted of intramuscular administration of 3.3 mg/kg flunixin meglumine (Finadyne, Schering-Plough Animal Health, Oss, The Netherlands), was set at score 4.

### 2.6. Statistical Analysis

Software G*Power 3.1 was used to determine the appropriate sample size. An “a priori” ANOVA (fixed-effects, omnibus, one-way) was conducted, with an effect size (f) of 0.45, significance level (α) of 0.05, power (1-β) of 0.80, and three groups. Statistical analysis was performed using SPSS 15.0 (IBM Company, Novegro-Tregarezzo, Italy). Shapiro–Wilk normality was applied. Data were reported as mean ± SD or median and range. Differences in physiological parameters, over time and between groups, were assessed by two-way ANOVA for repeated measures, followed by Bonferroni test. Differences along the timeline and between groups in stimulus response scores were compared using Friedman’s test. Inter-observer agreement for postoperative pain scores was assessed using Kendall’s concordance coefficient (W). SPSS automatically corrects the data. Statistical significance was set at *p* < 0.05.

## 3. Results

The total sample size was 66 subjects. The actual power was 0.8. Inter-observer agreement was high (W = 1). Data were not normally distributed. All selected swine completed the study. The trend of *f_R_*, HR, SAP, MAP, DAP, and SPO_2_ (98/100%) and T° (39.5°/38 °C) (for the last two parameters details not shown in Table 1) was adequate for the anesthetic and analgesic planes observed and normal for a patient under anesthesia. The duration of surgery was approximately 30 min.

In the LL group, significant differences were found in CPS scores at T15, T25, and T30 compared with T10 (*p* = 0.039, *p* = 0.006, and *p* = 0.006, respectively). No significant differences in CPS scores were observed in the LT group. In the TT group, significant differences were found in CPS scores at T15, T20, T25, and T30 compared with T10 (*p* = 0.006, *p* = 0.013, *p* = 0.006, and *p* = 0.013, respectively).

Comparing the CPS scores between the groups at each time point, significant differences were observed (*p* < 0.001). The TT group had higher scores than the LT group at T10 and T15 (*p* = 0.006 and *p* = 0.003). The LT group had lower scores than the LL group at T10, T15, T25, and T30 (*p* = 0.034, *p* < 0.001, *p* = 0.005 and *p* < 0.001). The LL group had lower scores than the TT group at T10 (*p* = 0.007) and higher scores than TT group at T30 (*p* = 0.017). All swine had CPS scores < 10 (Table 2). The postoperative pain score was ≤4 throughout the observation period. In the LL group, UPAPS showed variations along the timeline from R2 to R6 (*p* < 0.001).

In LT and TT groups, UPAPS did not show any variations along the timeline. Comparison between groups regarding UPAPS scores showed a significant difference between LL and LT (*p* < 0.001), and between LL and TT (*p* < 0.001), as LT and TT scores were lower than LL scores along the postoperative timeline (Table 3). No subject required rescue analgesia and no side effects, such as delirium, allergic reactions, hyperthermia, or hypothermia, were observed.

## 4. Discussion

The anesthetic protocol used in this study involved the intramuscular administration of a combination of tiletamine/zolazepam at 5 mg/kg and romifidine at 80 µg/kg, followed by the administration of lidocaine and tramadol along the incision lines and intraperitoneally. This protocol was able to provide adequate anesthesia and analgesia in pigs undergoing umbilical hernia repair, with no clinically demonstrable adverse effect. The number of legally authorized anesthetic and analgesic drugs for livestock, including swine, is limited. Therefore, it is necessary to suggest therapeutic alternatives to veterinarians designed to ensure the patient’s welfare, the surgeon’s comfort, and the safety of the staff.

Romifidine, telazol, and tramadol, although used in livestock such as cattle and horses, are not registered for these species. Romifidine is a 2-adrenergic (2-AA) agent which provides sedation and analgesia with a short period of hypertension and reflex compensatory bradycardia followed by hypotension. 2-Adrenergic agonists (2-AA) cause bradypnea, resulting in hypoxia, hypercapnia, increased urine output, and hyperglycemia [30,31,40,41,42,43,44,45,46,47]. The onset, duration, and depth of 2-AA-induced sedation can be influenced by stressful climatic environmental factors and the subject’s temperament [37,38,40,41,48,49].

To minimize the side effects and improve the therapeutic efficacy of 2-AAs, in clinical practice they are administered in association with opioids that act on the same membrane G-protein [30,42,43,44]. However, in cases of 2-AA-induced side effects, which can vary in intensity and duration, the effect can be antagonized by atipamezole, which may also reduce the analgesic efficacy of the co-administered opioid [45].

In recent studies, romifidine, which is approved for use in horses, has been administered to cattle in combination with tramadol [31,33]. The administration of romifidine at 0.02 mg/kg intramuscularly, combined with 1 mg/kg tramadol given slowly intravenously, provided adequate sedation and analgesia for minor surgery in standing cattle with minimal ataxia [31].

The administration of romifidine at a dose of 80 µg/kg intramuscularly, followed by tramadol at 1 mg/kg intravenously and lidocaine applied at the incision site, provided adequate sedation and analgesia for umbilical hernia surgery in cattle [13]. In horses, a combination of romifidine at 50 µg/kg intravenously and tramadol at a dose of 3 mg/kg, given slowly intravenously over 15 min, provided adequate sedation and analgesia for performing diagnostic tests with the patient in the upright position [30].

Telazol is a combination of tiletamine and zolazepam, approved for dogs and cats but also used in clinical practice in swine. Pharmacokinetic studies in swine have shown that tiletamine has a lower in vitro metabolic stability than zolazepam. These results suggest that tiletamine and zolazepam have different durations of action and therefore their therapeutic synergism is transient [46].

Tramadol is an analgesic that works with two different mechanisms. The first one is related to the non-opioid alpha-2 agonist and serotonergic activity, while the second one is related to the opioid effect. The latter is enhanced by the monoaminergic effect, which partially antagonizes the μ-opioid receptors (MORs) involved in the recognition of enkephalins and beta-endorphin [50]. Interaction with this receptor determines the inhibition of voltage-gated calcium channel required for the exocytosis of cellular vesicles in synaptic clefts, which controls neurotransmitter release by the presynaptic neuron [51]. On the other hand, MOR resistance in the postsynaptic neuron mediates the opening of potassium channels, resulting in hyperpolarization of the spinothalamic neuron and its reduced excitability [51]. This is explained by a reduction in the conduction of the painful stimulus to higher nerve centers due to the opiate effect, which silences the first-order neuron, deafens the second-order neuron, and inhibits serotonin and norepinephrine reuptake. The lower respiratory depression of tramadol compared with other opioid analgesics makes it optimal for the control of postoperative and chronic pain. These properties also make it a useful preanesthetic, facilitating intubation, reducing the dose of the anesthetic induction agent, and improving recovery. Tramadol, a compound with structural similarity to the centrally acting opioids codeine and morphine, has been shown to modulate pain via the spine and prevent transmission to the brain. Tramadol does not produce serious side effects such as respiratory and cardiovascular depression, tremors, muscle fasciculations, ataxia, or agitation [52]. However, the rapid achievement of high plasma concentrations can induce the ‘serotonin storm’ effect in some species [53].

In the present study, the combination of tiletamine-zolazepam, romifidine, lidocaine, and tramadol represents a well-balanced anesthetic protocol that provides sedation, analgesia, and muscle relaxation, and allows the patient to be placed in the dorsal recumbent position. The combination of tiletamine/zolazepam and romifidine has been used for the remote capture of wild boars, whose clinical monitoring revealed a favorable cardiovascular and respiratory homeostasis, possibly due to the short induction times during anesthesia [34,48,54,55].

The well-being of surgical patients depends primarily on the effective management of their response to the surgical stimulus. Achieving minimal or no response to the surgical stimulus allows the establishment of a stable anesthetic plan and improves surgeon comfort. Unfortunately, these crucial aspects are often overlooked in livestock, where surgery is often performed on the farm, using only local anesthetics, which can be ineffective [13,32].

Locoregional anesthesia represents the main anesthetic technique used in livestock, as almost all surgery is performed with the animal in a standing position, making local anesthetics essential. However, dorsal recumbency is also required for abdominal surgery such as umbilical hernia repair [13,32].

It is important to consider the reduced efficacy of local anesthetics in acidic environments, such as inflamed tissues [17]. Given the critical importance of this factor and recognizing that lidocaine is the most commonly used drug for livestock surgical procedures, it is necessary to have alternative drugs that provide local analgesia comparable to lidocaine. Tramadol may be a viable alternative in this case.

Pain can have several effects on the body’s homeostasis, including the release of catecholamines, leading to cardiovascular and circulatory stress manifested in tachycardia and hypertension, and the release of cortisol, which increases glucagon levels and decreases insulin levels. Pain also has catabolic effects, causing delayed healing of operative wounds and reduced mobility, resulting in pulmonary atelectasis, inadequate nutrition leading to hypotrophy of the intestinal villi, and excessive lipolysis with the release of ketone bodies [45,56].

The CPS scores showed that lidocaine had a faster onset of action than tramadol, which was more effective over time than lidocaine. In fact, the TT group had a higher CPS score at T10 compared to LL and LT (score 3/4 vs. 0/2). However, HR in the LL group, and SAP and DAP in the LT group, were significantly higher at T1 (baseline). The combination of lidocaine and tramadol was the most effective in terms of both onset and duration. However, CPS scores were below the established cut-off point in all groups and no subjects required rescue analgesia. Similar results were obtained in a study conducted in dogs undergoing ovariohysterectomy, in which different groups received lidocaine, tramadol, or a combination of both, intraperitoneally, for postoperative pain management [10].

The combination of lidocaine and tramadol produces a longer lasting anesthesia than lidocaine [57]. The administration of tramadol and lidocaine epidurally, especially for perineal surgery in cows and when the animal is required to maintain a standing position, results in pain relief and improved pain management, as observed in other studies [58]. The main limitation of the study is that only the analgesia achieved was considered. In fact, lidocaine and tramadol were used in patients undergoing general anesthesia obtained using romifidine and telazol, which provide sedation as well as analgesia. The combination of these drugs may have influenced the quality and duration of the analgesia. Moreover, the LT and TT groups obtained lower CPS and UPAPS scores over time than those of LL group. This is probably due to romifidine/telazol’s different interaction with lidocaine and tramadol. The synergism between opioids such as tramadol and alpha-2 agonists such as romifidine is well-known [13]. A further significant limitation is represented by the absence of a control group without lidocaine or tramadol. In our study, the UNESP-Botucatu pig composite acute pain scale (UPAPS) was used to assess postoperative pain. The scale was constructed by observing videos recorded in postoperative castrated swine. It consists of six items, each further divided into four subscales, with a maximum score of 18 points. Items assessed include posture, interaction and interest in surroundings, interaction with other animals, activity level, appetite, attention to affected area, and alternating pelvic limb elevation or support. In addition, the scale takes into account certain behaviors such as biting bars or objects, head position below the spine, and difficulty in overcoming obstacles. Each variable is assigned a score ranging from 0 to 3 [36]. However, assigning objective scores to swine can be difficult due to their stress sensitivity. Therefore, clinical and hematological assessments, such as evaluation of oxidative stress or cortisol levels, may not be reliable [59].

The limitations mentioned above lead us to the future perspective of also evaluating the sedation achieved with the drugs used in this study.

## 5. Conclusions

Tramadol and lidocaine, administered by infiltration of the surgical planes and intraperitoneally for the intraoperative and postoperative pain management in romifidine-telazol-anesthetized swine during umbilical hernia repair, provided adequate analgesia. Therefore, tramadol could also be used for pain management in livestock.

## Figures and Tables

**Table 1 animals-13-02905-t001:** Effect on physiological parameters of 5 mg/kg of tiletamine/zolazepam and 80 µg/kg of romifidine administered intramuscularly, followed by lidocaine (LL group), lidocaine and tramadol (LT group), and tramadol alone (TT group) by intraperitoneal and incisional line administration.

Groups	T1	T5	T10	T15	T20	T25	T30
HR (beats^min^)							
LL	146 ± 12 ^αβ^	61 ± 4 *^αβ^	104 ± 3 *^αβ^	105 ± 3 *^αβ^	93 ± 5 *^αβ^	86 ± 3 *^α^	89 ± 5 *^αβ^
LT	80 ± 3	76 ± 3 *	42 ± 3 *^δ^	44 ± 3 *^δ^	68 ± 5 *	67 ± 2 *	84 ± 2 ^δ^
TT	84 ± 10	78 ± 12	80 ± 3	78 ± 15	77 ± 11	69 ± 16	77 ± 3
*f_R_* (breaths^min^)							
LL	70 ± 3 ^αβ^	66 ± 4 ^αβ^	67 ± 4 ^αβ^	68 ± 3 ^αβ^	57 ± 3 *	52 ± 3 *^α^	63 ± 3 *^β^
LT	60 ± 3 ^δ^	57 ± 3 ^δ^	64 ± 2 *^δ^	63 ± 2 *^δ^	55 ± 8	63 ± 2 *^δ^	64 ± 3 *^δ^
TT	51 ± 4	50 ± 3	58 ± 3 *	44 ± 2 *	54 ± 3	50 ± 3	51 ± 2
SAP (mmHg)							
LL	123 ± 2 ^αβ^	161 ± 3 *^αβ^	141 ± 3 *^αβ^	148 ± 12 *	135 ± 10 ^α^	116 ± 2 *^αβ^	167 ± 2 *^β^
LT	164 ± 2 ^δ^	166 ± 2	158 ± 3 *^δ^	157 ± 5 *	185 ± 19 ^δ^	152 ± 3 *^δ^	170 ± 25 ^δ^
TT	140 ± 2	170 ± 4 *	163 ± 3 *	158 ± 9 *	137 ± 4	132 ± 3 *	133 ± 3 *
MAP (mmHg)							
LL	111 ± 7 ^β^	102 ± 4 *^αβ^	107 ± 4 ^α^	111 ± 8 ^αβ^	104 ± 14 ^αβ^	87 ± 4 *^α^	114 ± 4 ^β^
LT	109 ± 3 ^δ^	108 ± 3	93 ± 2 *^δ^	94 ± 4 *^δ^	116 ± 12 ^δ^	98 ± 2 *^δ^	113 ± 14 ^δ^
TT	97 ± 6	109 ± 9 *	108 ± 2 *	103 ± 10	92 ± 7 *	88 ± 8 *	92 ± 3 *
DAP (mmHg)							
LL	65 ± 2 ^αβ^	82 ± 2 *^β^	72 ± 3 *^αβ^	77 ± 7 *	70 ± 8 ^α^	59 ± 3 *^αβ^	84 ± 2 *^β^
LT	82 ± 3 ^δ^	83 ± 2 ^δ^	79 ± 1 *^δ^	79 ± 3 *^δ^	93 ± 10	76 ± 2 *	85 ± 13 ^δ^
TT	70 ± 3	85 ± 2 *	81 ± 2 *	80 ± 2 *	70 ± 2	66 ± 1 *	66 ± 2 *

HR = heart rate; *f_R_* = respiratory frequency; SAP = non-invasive systolic arterial blood pressure; MAP = non-invasive mean arterial blood pressure; DAP = non-invasive diastolic arterial blood pressure; T1 (after sternal recumbency); T5 (five minutes after dorsal recumbency); T10 (skin incision); T15 (muscle plane incision); T20 (herniary sac opening and herniorrhaphy); T25 (muscle plane suture), and T30 (skin suture). * Differences along the timeline; ^α^ global differences between group LL and group LT; ^β^ global differences between group LL and group TT; ^δ^ global differences between group LT and group TT. Data were reported as mean ± SD.

**Table 2 animals-13-02905-t002:** The intraoperative surgical stimulus response of 5 mg/kg of tiletamine/zolazepam and 80 µg/kg of romifidine, administered intramuscularly, followed by lidocaine (LL group), lidocaine and tramadol (LT group), and tramadol alone (TT group), administered intraperitoneally and into incisional lines, was assessed by cumulative pain score (CPS).

CPS Score	T10	T15	T20	T25	T30
LL	1 (1/2) ^β^	3 (1/4) *	1 (0/3)	0 (0/0) *	3 (3/3) *^β^
LT	1 (0/1) ^αδ^	1 (1/1) ^α^	1 (0/2)	1 (0/1) ^α^	1 (1/4) ^α^
TT	4 (3/4) ^δβ^	2 (2/2) *^δ^	1 (0/5) *	0 (0/2) *	0 (0/5) *

Numeric score ranging from 0 to 4 assigned based on the percentage variation from baseline values of each measured physiological parameter (*f_R_*, HR, and SAP) at T10 (skin incision), T15 (muscle plane incision), T20 (herniary sac opening and herniorrhaphy), T25 (muscle plane suture), and T30 (skin suture), according to the following scheme: 0 = ≤0%, 1 = >0% but ≤10%, 2 = >10% but ≤20%, 3 = >20% but ≤30%, and 4 = >30%. * Difference along the timeline; ^α^ global differences between group LL and group LT; ^β^ global differences between group LL and group TT; ^δ^ global differences between group LT and group TT. Data were reported as median and range.

**Table 3 animals-13-02905-t003:** Effect of tiletamine/zolazepam at 5 mg/kg and romifidine at 80 µg/kg, administered intramuscularly, followed by lidocaine (LL group), lidocaine and tramadol (LT group), and tramadol alone (TT group), administered intraperitoneally and into incision lines, on postoperative pain evaluated by the UNESP-Botucatu pig composite acute pain scale (UPAPS).

UPAPS Score	R0	R1	R2	R3	R4	R5	R6
LL	0 (0/0)	0 (0/0)	1 (1/3) *^αβ^	2 (1/3) *^αβ^	3 (3/3) *^αβ^	3 (3/4) *^αβ^	3 (3/4) *^αβ^
LT	0 (0/0)	0 (0/0)	0 (0/0)	0 (0/0)	0 (1/1)	0 (1/1)	0 (1/1)
TT	0 (0/0)	0 (0/0)	0 (0/0)	0 (0/0)	0 (0/0)	0 (0/0)	0 (0/0)

(R0) after a standing position was adapted; (R1, R2, R3, R4, R5, and R6) each of the subsequent six hours. * Difference along the timeline; ^α^ global differences between group LL and group LT; ^β^ global differences between group LL and group TT; Data were reported as mean ± SD or median and range.

## Data Availability

The data presented in this study are available on request from the corresponding author.

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
