# Peer review of "Tramadol vs. Lidocaine Administered Intraperitoneally and in Incisional Lines for the Intraoperative and Postoperative Pain Management of Romifidine-Telazol-Anesthetized Swine Undergoing Umbilical Hernia Repair"

_animals, 2023, doi:10.3390/ani13182905_

Round 1
Reviewer 1 Report
Overall, a good study. However, I find it curious that such an old version of SPSS has been used. On the other hand, I missed the values of the statistics (in this case, the F) as well as the significance.
at some points the reading is not fluent. Apart from that, nothing to highlight.
Author Response
REVIEW 1
Dear and esteemed reviewer, thank you for your precious suggestions, they have been very useful for me to improve the manuscript
Comments and Suggestions for Authors
Overall, a good study. However, I find it curious that such an old version of SPSS has been used. On the other hand, I missed the values of the statistics (in this case, the F) as well as the significance.
Comments on the Quality of English Language
at some points the reading is not fluent. Apart from that, nothing to highlight. Answers
I improved the English and made the significance clearer.

Reviewer 2 Report
The use of local anesthetics and opioids as a multimodal analgesia strategy is an increasingly common technique in animals, where pigs are no exception, especially in incisional blocks or epidurals. The main strength of this manuscript is that it presents a comparative study of the analgesic efficacy of tramadol and lidocaine by incisional and intraperitoneal routes in the surgical model of umbilical hernia repair in pigs. As reviewer, I must emphasize that this study provides an effective alternative to pain acute management in pigs.
However, some points must be addressed to achieve publication quality. I have left some comments hoping that they can help the authors.
General comments
L35: please add a conclusion to the abstract.
L58-59: please complement what is expressed in these lines, since lidocaine has a more complex mechanism of action. In addition, it has other pharmacological properties, including anti-inflammatory. I leave some references with the hope of helping the authors:
10.1111/j.1399-6576.1996.tb04434.x
10.1007/s40265-018-0955-x
10.1016/j.bja.2019.06.014
10.1016/j.bpa.2017.05.005
L67 and 70: please add a reference. I suggest the authors include the following:
10.5455/javar.2021.h529
L83-85: In general, materials and methods should be divided into sections. I suggest to the authors that the cited lines form section 2.1 and that it be called animals. At the end of it, please add the exclusion criteria considered in your study.
L86-95: I suggest these lines be included in section 2.2 called anesthetic procedure.
L91: Was the fluid therapy administered through an intravenous pump fluid? If so, please clarify.
L96-102 and L111-117: I suggest these lines be included in section 2.3 called experimental design.
L97-100: the authors specify that the animals received the treatments by infiltration and by intraperitoneal route, but, of the doses indicated in lines 92-93, how much was administered by each route? please clarify.
L103-110: I suggest these lines be included in section 2.4 called surgical procedure.
L112: Was the MAP calculated by formula adjustment or was it taken directly from the monitor?
L113-114: Is the body temperature parameter was not monitored? In addition, I suggest to authors that when referring to equipment, devices, and drugs, they do so by mentioning their name, model, brand or laboratory, and country of origin. Just like they did on the L88.
L118-131: I suggest these lines be included in section 2.5, called Evaluation of intraoperative nociception and postoperative pain.
L125: Please indicate the route of administration and that lidocaine was used as intraoperative rescue analgesia.
L132-139: I suggest these lines be included in section 2.6 called statistical analysis. In this paragraph, it remains to indicate the statistical analysis applied to the cardiorespiratory variables.
Table 1: To improve the understanding of the results expressed in this table, I suggest the authors switch to mean and standard deviation. If the deviations are large by the calculated coefficient of variation, then they could express the cardiorespiratory parameter results as mean and standard error of the mean. This change would also make an analysis of variance for repeated measures a better option than Friedman’s analysis.
Table 2: the symbol & is not indicated at the bottom of the table.
L228: In the present study, the combination of….
L229: “represents”
L240: please add a reference.
L256: This document could help support what is described in this paragraph. In addition, to explain some effects of the drugs used:
10.14202/vetworld.2021.393-404
Discussion: In general, a greater depth is required to explain the analgesic and cardiorespiratory effects of both lidocaine and tramadol, especially to highlight the advantages of using this atypical opioid. I suggest the authors search for some studies to compare the results obtained in their experiment. I leave some suggestions with the hope of helping you:
10.1213/ANE.0000000000003573
10.1093/jat/bkab044
10.1111/jvp.12133
10.1136/vr.c4458
10.1111/j.1467-2995.2009.00496.x
Although it is about other species, these articles could be useful to highlight the advantages of the tramadol-lidocaine combination:
10.1080/00480169.2012.696576
10.3390/ani13071145
10.1111/j.1365-2885.2010.01158.x
L275: Please add a discussion of study limitations and future directions for your research.
Minor editing of English language required
Author Response
REVIEW 2
Comments and Suggestions for Authors
The use of local anesthetics and opioids as a multimodal analgesia strategy is an increasingly common technique in animals, where pigs are no exception, especially in incisional blocks or epidurals. The main strength of this manuscript is that it presents a comparative study of the analgesic efficacy of tramadol and lidocaine by incisional and intraperitoneal routes in the surgical model of umbilical hernia repair in pigs. As reviewer, I must emphasize that this study provides an effective alternative to pain acute management in pigs.
However, some points must be addressed to achieve publication quality. I have left some comments hoping that they can help the authors.
General comments
Dear and esteemed reviewer, thank you for your precious suggestions, they have been very useful for me to improve the manuscript
L35: please add a conclusion to the abstract.
I added the conclusions in the abstract and deleted some words
L58-59: please complement what is expressed in these lines, since lidocaine has a more complex mechanism of action. In addition, it has other pharmacological properties, including anti-inflammatory. I leave some references with the hope of helping the authors:
10.1111/j.1399-6576.1996.tb04434.x
10.1007/s40265-018-0955-x
10.1016/j.bja.2019.06.014
10.1016/j.bpa.2017.05.005
Done
L67 and 70: please add a reference. I suggest the authors include the following:
10.5455/javar.2021.h529
Done
L83-85: In general, materials and methods should be divided into sections. I suggest to the authors that the cited lines form section 2.1 and that it be called animals. At the end of it, please add the exclusion criteria considered in your stud
Done
L86-95: I suggest these lines be included in section 2.2 called anesthetic procedure.
Done
L91: Was the fluid therapy administered through an intravenous pump fluid? If so, please clarify.
Done
L96-102 and L111-117: I suggest these lines be included in section 2.3 called experimental design.
Done
L97-100: the authors specify that the animals received the treatments by infiltration and by intraperitoneal route, but, of the doses indicated in lines 92-93, how much was administered by each route? please clarify.
Done
L103-110: I suggest these lines be included in section 2.4 called surgical procedure.
Done
L112: Was the MAP calculated by formula adjustment or was it taken directly from the monitor?
Directly from the monitor
L113-114: Is the body temperature parameter was not monitored? In addition, I suggest to authors that when referring to equipment, devices, and drugs, they do so by mentioning their name, model, brand or laboratory, and country of origin. Just like they did on the L88.
I added name, model, brand or laboratory, and country of origin, and brief details on the temperature
L118-131: I suggest these lines be included in section 2.5, called Evaluation of intraoperative nociception and postoperative pain.
Done
L125: Please indicate the route of administration and that lidocaine was used as intraoperative rescue analgesia.
Done
L132-139: I suggest these lines be included in section 2.6 called statistical analysis. In this paragraph, it remains to indicate the statistical analysis applied to the cardiorespiratory variables.
Done
Table 1: To improve the understanding of the results expressed in this table, I suggest the authors switch to mean and standard deviation. If the deviations are large by the calculated coefficient of variation, then they could express the cardiorespiratory parameter results as mean and standard error of the mean. This change would also make an analysis of variance for repeated measures a better option than Friedman’s analysis.
Done
Table 2: the symbol & is not indicated at the bottom of the table.
We corrected it
L228: In the present study, the combination of….
Done
L229: “represents”
Done
L240: please add a reference.
Done
L256: This document could help support what is described in this paragraph. In addition, to explain some effects of the drugs used:
10.14202/vetworld.2021.393-404
DONE
Discussion: In general, a greater depth is required to explain the analgesic and cardiorespiratory effects of both lidocaine and tramadol, especially to highlight the advantages of using this atypical opioid. I suggest the authors search for some studies to compare the results obtained in their experiment. I leave some suggestions with the hope of helping you:
10.1213/ANE.0000000000003573
10.1093/jat/bkab044
10.1111/jvp.12133
10.1136/vr.c4458
10.1111/j.1467-2995.2009.00496.x
Although it is about other species, these articles could be useful to highlight the advantages of the tramadol-lidocaine combination:
10.1080/00480169.2012.696576
10.3390/ani13071145
10.1111/j.1365-2885.2010.01158.x
Done
L275: Please add a discussion of study limitations and future directions for your research.
Done
Comments on the Quality of English Language
Minor editing of English language required
Done
Reviewer 3 Report
The article is up-to-date, interesting and contributing. The use of local anesthetic techniques in animals is important especially from the aspect of postoperative analgesia. I consider the new knowledge about the use of tramadol in pigs to be very beneficial.
Abstract
L 23 – Change "80 µ/kg" to "80 µg/kg".
L 23 – It is not stated how many animals each group contained.
L 30 – I recommend replacing “respiratory rate (RR)" with "respiratory frequency (fR)" (in the entire manuscript).
Keywords
L 35 – the words in the title and keywords are repeated.
Introduction
L 40-45 – I recommend removing the paragraph about umbilical hernia, the manuscript is focused on locoregional anesthesia and analgesia.
L 53 – Change "gen-eral" to "general".
L 54 – I recommend replacing “dorsal decubitus" with "dorsal recumbency" (in the entire manuscript).
At the end of the Introduction, a hypothesis should be stated.
Materials and Methods
L 87-89 – tiletamine/zolazepam and romifidine were administered in one syringe or separately?
L 90 – "After achieving anaesthesia" - what does it mean? please describe more precisely how the state of achieving anesthesia was assessed.
L 90 – "swine were cannulated into the jugular vein" - were the pigs really cannulated into the jugular vein? This is not usual in normal practice.
L 92 – Change "ml/kg/h" to "ml/kg/hour".
L 94 – "the volumes of each drug were increased to 40 ml" – In the LL/TT group, lidocaine/tramadol was diluted with saline to 40 milliliters? how many milliliters was given for the incisional and how many for the intraperitoneal block?
Was the size of the hernia recorded? Were all the hernias the same size? Administration of the identical volume of drugs may produce a different response in hernias of different sizes, which can significantly affect the results.
L 111 – I recommend replacing “systolic, diastolic, and mean blood pressure" with " systolic, diastolic, and mean arterial pressure" (in the entire manuscript).
L 112 – for "hemoglobin saturation" add "of oxygen".
L 111-112 – describe more precisely how HR, SAP, DAP, MAP and SpO2 was measured.
L 115-116 – specify and add what T10, T15, T20, T25 and T30 mean. Are these times after dorsal recumbency?
L 118 – “Intraoperative surgical stimulus” – what does it mean? Please, describe exactly how to invoke it. Was it a skin incision? Reaction to peritoneum manipulation etc.?
L 124-125 – How was the lidocaine administered?
L 126-131 – When was postoperative pain assessed? Please complete the times.
L 127 – "by three observers” – all at the same time or did all three participate in the assessment of all animals together?
L 132 – according to which parameter the sample size selection was calculated?
On L 156, you state the variation, but it is not stated how it was calculated.
Exclusion criteria are not defined.
Results
I recommend describing differences "within groups" and "between groups" (throughout Results).
I recommend replacing "p = 0.000" with "p < 0.001" (throughout Results).
L 160 – "side effects" – how were the side effects assessed?
Tables
The legend to the tables should provide all the data so that the tables is comprehensible, and if it will be published separately from the main text.
What do the numbers mean? Mean/minimum/maximum? I recommend writing in the format mean (minimum – maximum).
HR in the LL group and SAP and DAP in the LT group are significantly higher in T1 (baseline). Why? I recommend commenting or explaining in the discussion.
Discussion
The manuscript is focused on the effects of lidocaine and tramadol. I therefore recommend focusing the discussion on these drugs, not on romifidine or zolazepam/tiletamine. I therefore recommend reducing or removing paragraphs on L 200-212, L 223-227 and L 231-240.
Conclusions
The differences between LL, LT and TT described in Results are not mentioned here.
The manuscript is beneficial for the professional public as well as for practice. In view of the above comments, however, I do not recommend accepting it in the present form. I recommend correcting and supplementing the manuscript and sending it for repeated review.
Author Response
REVIEW 3
Comments and Suggestions for Authors
The article is up-to-date, interesting and contributing. The use of local anesthetic techniques in animals is important especially from the aspect of postoperative analgesia. I consider the new knowledge about the use of tramadol in pigs to be very beneficial.
Dear and esteemed reviewer, thank you for your precious suggestions, they have been very useful for me to improve the manuscript
Abstract
L 23 – Change "80 µ/kg" to "80 µg/kg".
Done
L 23 – It is not stated how many animals each group contained.
Done
L 30 – I recommend replacing “respiratory rate (RR)" with "respiratory frequency (fR)" (in the entire manuscript).
Done
Keywords
L 35 – the words in the title and keywords are repeated.
Done
Introduction
L 40-45 – I recommend removing the paragraph about umbilical hernia, the manuscript is focused on locoregional anesthesia and analgesia.
L 53 – Change "gen-eral" to "general".
Done
L 54 – I recommend replacing “dorsal decubitus" with "dorsal recumbency" (in the entire manuscript).
Done
At the end of the Introduction, a hypothesis should be stated.
Done
Materials and Methods
L 87-89 – tiletamine/zolazepam and romifidine were administered in one syringe or separately?
Tiletamine/zolazepam and romifidine were administered in one syringe
L 90 – "After achieving anaesthesia" - what does it mean? please describe more precisely how the state of achieving anesthesia was assessed.
Done
L 90 – "swine were cannulated into the jugular vein" - were the pigs really cannulated into the jugular vein? This is not usual in normal practice.
Yes, they were actually cannulated in the jugular as the limbs were bound by gauze.
L 92 – Change "ml/kg/h" to "ml/kg/hour".
Done
L 94 – "the volumes of each drug were increased to 40 ml" – In the LL/TT group, lidocaine/tramadol was diluted with saline to 40 milliliters? how many milliliters was given for the incisional and how many for the intraperitoneal block?
I specified in the text
Was the size of the hernia recorded? Were all the hernias the same size? Administration of the identical volume of drugs may produce a different response in hernias of different sizes, which can significantly affect the results.
I specified in the text
L 111 – I recommend replacing “systolic, diastolic, and mean blood pressure" with " systolic, diastolic, and mean arterial pressure" (in the entire manuscript).
DONE
L 112 – for "hemoglobin saturation" add "of oxygen".
DONE
L 111-112 – describe more precisely how HR, SAP, DAP, MAP and SpO2 was measured.
Done
L 115-116 – specify and add what T10, T15, T20, T25 and T30 mean. Are these times after dorsal recumbency?
They correspond to precise surgical moments to have the same surgical stimulus
L 118 – “Intraoperative surgical stimulus” – what does it mean? Please, describe exactly how to invoke it. Was it a skin incision? Reaction to peritoneum manipulation etc.?
they were incisions, manipulations and sutures of the various surgical planes
L 124-125 – How was the lidocaine administered?
Done
L 126-131 – When was postoperative pain assessed? Please complete the times.
Done
L 127 – "by three observers” – all at the same time or did all three participate in the assessment of all animals together?
Done
L 132 – according to which parameter the sample size selection was calculated?
The inclusion criterion was the presence of 3-5 cm diameters umbilical hernia inserted in materials and methods
On L 156, you state the variation, but it is not stated how it was calculated.
DONE
Exclusion criteria are not defined.
DONE
Results
I recommend describing differences "within groups" and "between groups" (throughout Results).
DONE
I recommend replacing "p = 0.000" with "p < 0.001" (throughout Results).
we have stated the actual p values given by SPSS
L 160 – "side effects" – how were the side effects assessed?
Delirium, allergic reactions, hyperthermia, hypothermia.
DONE
Tables
The legend to the tables should provide all the data so that the tables is comprehensible, and if it will be published separately from the main text.
DONE
What do the numbers mean? Mean/minimum/maximum? I recommend writing in the format mean (minimum – maximum).
DONE
HR in the LL group and SAP and DAP in the LT group are significantly higher in T1 (baseline). Why? I recommend commenting or explaining in the discussion.
Discussion
The manuscript is focused on the effects of lidocaine and tramadol. I therefore recommend focusing the discussion on these drugs, not on romifidine or zolazepam/tiletamine. I therefore recommend reducing or removing paragraphs on L 200-212, L 223-227 and L 231-240.
Done
Conclusions
The differences between LL, LT and TT described in Results are not mentioned here.
Done
The manuscript is beneficial for the professional public as well as for practice. In view of the above comments, however, I do not recommend accepting it in the present form. I recommend correcting and supplementing the manuscript and sending it for repeated review.
Done
Reviewer 4 Report
Dear authors,
it is very important to do such studies as we have to improve animal welfare regarding food producing animals. In order to be able to really judge your paper in a first step the M+M and result section has to be rewritten to make it clearer, more concise and to the point. As almost all drugs used in this study are not allowed in food producing animals, this aspect should also be added to the manuscript. Please find below more specific comments.
Introduction:
Phrase on line 43 -45 is irrelevant
Line 56 as far as I'm aware in europe in most countries only procaine is allowed in food producing animals- so I'm not certain if this statement is true- and it is not relevant for the study.
Phrase 70-72 is irrelevant to the study and should be deleted.
M+M
The whole timing of the study is very important and has to be described in detail (otherwise it is not possible for me to judge the tremendously different Numbers you present in table 1 (did you apply test drugs before they wnt into sternal recombency?):
The application of the test drugs has to be described much better: when exactly were they administered, how much to each location? did you give a total of 40 ml per pig- or in the LT group 80 ml? or on all groups 80 ml??Were the surgeons always the same?
Numeric score line 119 and following: which score were animals reaching 10, 20 or 30 given? they way you describe it only 11, 21, and 31 are clearly defined.
For which parameter did you estimate the sample size- and which criteria did you judge significant- please describe in more detail.
Line 124 -where did you administer the lidocaine bolus
Results:
how long did the surgery take
how often did you have to give lidocaine rescue bolus intraoperatively? Did you exclude those pigs from study?
What do the 3 numbers each in table 1 mean? Which number did you take to judge the score? You gave the score during anaesthesia or retrospectively? If at T1 you have 90/61/93 and at T5 83/43/86-which score would that pig get-and so on for the following time points?
Postoperative pain scores:
How long did the pigs take to become standing?Big difference between individual pig or even groups???
Did you start with T6 then and scored every hour- describe in more detail and more precise! The timing has a big influence as you have several drugs that they got with various durations of action... It would be better to give intraoperative timepoints a different number from post op points- post op ev R1, R2, R3...
Discussion
will be judged carefully once results have been cleared-BUT think what is relevant to the paper and delete the remaining discussion that is not relevant. Relevant is the duration of action of the used drugs (all of them also the once you used for anaesthesia), the dose rates of the study drugs and the expected effects, do you think the effects you saw were local or more a result of absorption of study drugs etc...
should be corrected by an english native
Author Response
REVIEW 4
Comments and Suggestions for Authors
Dear authors,
it is very important to do such studies as we have to improve animal welfare regarding food producing animals. In order to be able to really judge your paper in a first step the M+M and result section has to be rewritten to make it clearer, more concise and to the point. As almost all drugs used in this study are not allowed in food producing animals, this aspect should also be added to the manuscript. Please find below more specific comments.
Dear and esteemed reviewer, thank you for your precious suggestions, they have been very useful for me to improve the manuscript
Introduction:
Phrase on line 43 -45 is irrelevant
Line 56 as far as I'm aware in europe in most countries only procaine is allowed in food producing animals- so I'm not certain if this statement is true- and it is not relevant for the study.
Phrase 70-72 is irrelevant to the study and should be deleted.
M+M
The whole timing of the study is very important and has to be described in detail (otherwise it is not possible for me to judge the tremendously different Numbers you present in table 1 (did you apply test drugs before they wnt into sternal recombency?):
Done
The application of the test drugs has to be described much better: when exactly were they administered, how much to each location? did you give a total of 40 ml per pig- or in the LT group 80 ml? or on all groups 80 ml??Were the surgeons always the same?
Done
Numeric score line 119 and following: which score were animals reaching 10, 20 or 30 given? they way you describe it only 11, 21, and 31 are clearly defined.
Response to surgical stimulus was assessed using a cumulative pain scale (CPS) A
numeric score of 0 to 4 was assigned based on the percentage variation in RR, HR, and SAP
as follows: 0 = variation 0%; 1 = variation 10%; 2 = variation > 10% but 20%; 3 = variation > 20% but 30%; 4 = variation > 30%. The sum of the scores obtained for each parameter was response to surgical stimulus. [13 16,26,31].
For which parameter did you estimate the sample size- and which criteria did you judge significant- please describe in more detail.
Done
Line 124 -where did you administer the lidocaine bolus
I specified
Results:
how long did the surgery take
DONE
how often did you have to give lidocaine rescue bolus intraoperatively? Did you exclude those pigs from study?
It was not necessary to administer the rescue bolus
What do the 3 numbers each in table 1 mean? Which number did you take to judge the score? You gave the score during anaesthesia or retrospectively? If at T1 you have 90/61/93 and at T5 83/43/86-which score would that pig get-and so on for the following time points?
scores were assigned on raw HR, RR, and SAP data for each subject at each time point. The sum of the three scores gave the total score for each subject and for each single moment. We thus obtained 3 excel folders named CPSLL, CPSLT, CPSTT. Using SPSS we made a descriptive statistic, expressing the scores with median and range (as per statistical rule).
the score was assigned retrospectively, however any rescue bolus would have been administered with the aid of instrumental clinical monitoring.
Postoperative pain scores:
How long did the pigs take to become standing? Big difference between individual pig or even groups???
The pigs reached the standing position 20-30 minutes after the end of the surgery. However, the assessment of sedation deriving from the anesthetic protocol used will be the subject of a further study.
Did you start with T6 then and scored every hour- describe in more detail and more precise! The timing has a big influence as you have several drugs that they got with various durations of action... It would be better to give intraoperative timepoints a different number from post op points- post op ev R1, R2, R3...
DONE
Discussion
will be judged carefully once results have been cleared-BUT think what is relevant to the paper and delete the remaining discussion that is not relevant. Relevant is the duration of action of the used drugs (all of them also the once you used for anaesthesia), the dose rates of the study drugs and the expected effects, do you think the effects you saw were local or more a result of absorption of study drugs etc...
OK
Comments on the Quality of English Language
should be corrected by an english native

Round 2
Reviewer 2 Report
I thank the authors for considering my comments in the first revision of their manuscript. It seems to me that the article has improved substantially, so I am convinced that if this paper is published, readers will have an important contribution to favor the development of veterinary medicine, particularly in the field of anesthesia and pain management in swine.
However, in the new manuscript, I have found some aspects that require the authors' attention before publishing the article.
L62: please express as L/Kg
L68: please add a reference
L78: please add a reference
L83: “instamine”
L100: please add a reference
L103: please add a reference
L163: please indicate the cuff gauge
L214-215: Review Table 2 and you will see that the differences between LT and LL are found at T15 and T30.
L218: UBP or UPAPS.
Tables 2 and 3: indicate at the bottom of the table, the meaning of the values that are between parentheses.
Discussion
L272-294 that were added contains a correct explanation but these lines require relocation to L316 so that the discussion is organized and the reader's understanding is improved.
L278: please add a reference.
Finally, Table 2 mentions a significant difference in the TT group at T10 that requires further explanation, which the authors could find in the pharmacokinetic behavior of tramadol (particularly in the Cmax and Tmax values). This explanation can be added on the L346.
Author Response
REVIEW 2
Comments and Suggestions for Authors
The use of local anesthetics and opioids as a multimodal analgesia strategy is an increasingly common technique in animals, where pigs are no exception, especially in incisional blocks or epidurals. The main strength of this manuscript is that it presents a comparative study of the analgesic efficacy of tramadol and lidocaine by incisional and intraperitoneal routes in the surgical model of umbilical hernia repair in pigs. As reviewer, I must emphasize that this study provides an effective alternative to pain acute management in pigs.
However, some points must be addressed to achieve publication quality. I have left some comments hoping that they can help the authors.
General comments
Dear and esteemed reviewer, thank you for your precious suggestions, they have been very useful for me to improve the manuscript
L35: please add a conclusion to the abstract.
I added the conclusions in the abstract and deleted some words
L58-59: please complement what is expressed in these lines, since lidocaine has a more complex mechanism of action. In addition, it has other pharmacological properties, including anti-inflammatory. I leave some references with the hope of helping the authors:
10.1111/j.1399-6576.1996.tb04434.x
10.1007/s40265-018-0955-x
10.1016/j.bja.2019.06.014
10.1016/j.bpa.2017.05.005
Done
L67 and 70: please add a reference. I suggest the authors include the following:
10.5455/javar.2021.h529
Done
L83-85: In general, materials and methods should be divided into sections. I suggest to the authors that the cited lines form section 2.1 and that it be called animals. At the end of it, please add the exclusion criteria considered in your stud
Done
L86-95: I suggest these lines be included in section 2.2 called anesthetic procedure.
Done
L91: Was the fluid therapy administered through an intravenous pump fluid? If so, please clarify.
Done
L96-102 and L111-117: I suggest these lines be included in section 2.3 called experimental design.
Done
L97-100: the authors specify that the animals received the treatments by infiltration and by intraperitoneal route, but, of the doses indicated in lines 92-93, how much was administered by each route? please clarify.
Done
L103-110: I suggest these lines be included in section 2.4 called surgical procedure.
Done
L112: Was the MAP calculated by formula adjustment or was it taken directly from the monitor?
Directly from the monitor
L113-114: Is the body temperature parameter was not monitored? In addition, I suggest to authors that when referring to equipment, devices, and drugs, they do so by mentioning their name, model, brand or laboratory, and country of origin. Just like they did on the L88.
I added name, model, brand or laboratory, and country of origin, and brief details on the temperature
L118-131: I suggest these lines be included in section 2.5, called Evaluation of intraoperative nociception and postoperative pain.
Done
L125: Please indicate the route of administration and that lidocaine was used as intraoperative rescue analgesia.
Done
L132-139: I suggest these lines be included in section 2.6 called statistical analysis. In this paragraph, it remains to indicate the statistical analysis applied to the cardiorespiratory variables.
Done
Table 1: To improve the understanding of the results expressed in this table, I suggest the authors switch to mean and standard deviation. If the deviations are large by the calculated coefficient of variation, then they could express the cardiorespiratory parameter results as mean and standard error of the mean. This change would also make an analysis of variance for repeated measures a better option than Friedman’s analysis.
Done
Table 2: the symbol & is not indicated at the bottom of the table.
We corrected it
L228: In the present study, the combination of….
Done
L229: “represents”
Done
L240: please add a reference.
Done
L256: This document could help support what is described in this paragraph. In addition, to explain some effects of the drugs used:
10.14202/vetworld.2021.393-404
DONE
Discussion: In general, a greater depth is required to explain the analgesic and cardiorespiratory effects of both lidocaine and tramadol, especially to highlight the advantages of using this atypical opioid. I suggest the authors search for some studies to compare the results obtained in their experiment. I leave some suggestions with the hope of helping you:
10.1213/ANE.0000000000003573
10.1093/jat/bkab044
10.1111/jvp.12133
10.1136/vr.c4458
10.1111/j.1467-2995.2009.00496.x
Although it is about other species, these articles could be useful to highlight the advantages of the tramadol-lidocaine combination:
10.1080/00480169.2012.696576
10.3390/ani13071145
10.1111/j.1365-2885.2010.01158.x
Done
L275: Please add a discussion of study limitations and future directions for your research.
Done
Comments on the Quality of English Language
Minor editing of English language required
Done
I thank the authors for considering my comments in the first revision of their manuscript. It seems to me that the article has improved substantially, so I am convinced that if this paper is published, readers will have an important contribution to favor the development of veterinary medicine, particularly in the field of anesthesia and pain management in swine.
However, in the new manuscript, I have found some aspects that require the authors' attention before publishing the article.
L62: please express as L/Kg
DONE
L68: please add a reference
DONE
L78: please add a reference
DONE
L83: “instamine”
DONE
L100: please add a reference
DONE
L103: please add a reference
DONE
L163: please indicate the cuff gauge
DONE
L214-215: Review Table 2 and you will see that the differences between LT and LL are found at T15 and T30.
DONE
L218: UBP or UPAPS.
UPAPS
Tables 2 and 3: indicate at the bottom of the table, the meaning of the values that are between parentheses.
DONE
Discussion
L272-294 that were added contains a correct explanation but these lines require relocation to L316 so that the discussion is organized and the reader's understanding is improved.
DONE
L278: please add a reference.
DONE
Finally, Table 2 mentions a significant difference in the TT group at T10 that requires further explanation, which the authors could find in the pharmacokinetic behavior of tramadol (particularly in the Cmax and Tmax values). This explanation can be added on the L346.
DONE
Reviewer 3 Report
I thank the authors for their careful proofreading and answers to my comments. Unfortunately, I still can't find or understand the following comments:
Keywords
L 37 – the words in the title and keywords are repeated.
Introduction
At the end of the Introduction, I still do not find a stated hypothesis, what result the authors expect from the presented study.
L 62 + entire manuscript – I recommend unifying liters and milliliters - "l, ml" or "L, mL" (according to the journal's requirements)
L 81 – replace "magrophagic" with "macrophagic"
Materials and Methods
L 153, 156 – I recommend unifying information on suture materials according to the journals requirements (formula, trade name, manufacturer, city, state)
L 162 – the HR measurement method is not specified, please add it
L 168-169 – T10, T15, T20, T25, T30 are times after dorsal recumbency? Please add to each item.
Results
I recommend describing differences "within groups" and "between groups" (throughout Results and Tables).
If p = 0.000 and lower, I recommend writing p < 0.001 (in the entire manuscript).
Tables
What does the data mean? Mean ± SD? Median/minimum/maximum? Please indicate the values for each table.
HR in the LL group and SAP and DAP in the LT group are significantly higher in T1 (baseline). Why? I recommend commenting or explaining in the discussion.
The manuscript is beneficial for the professional public as well as for practice. In view of the above comments, I recommend accepting it after minor corrections.
Author Response
REVIEW 3
Comments and Suggestions for Authors
The article is up-to-date, interesting and contributing. The use of local anesthetic techniques in animals is important especially from the aspect of postoperative analgesia. I consider the new knowledge about the use of tramadol in pigs to be very beneficial.
Dear and esteemed reviewer, thank you for your precious suggestions, they have been very useful for me to improve the manuscript
Abstract
L 23 – Change "80 µ/kg" to "80 µg/kg".
Done
L 23 – It is not stated how many animals each group contained.
Done
L 30 – I recommend replacing “respiratory rate (RR)" with "respiratory frequency (fR)" (in the entire manuscript).
Done
Keywords
L 35 – the words in the title and keywords are repeated.
Done
Introduction
L 40-45 – I recommend removing the paragraph about umbilical hernia, the manuscript is focused on locoregional anesthesia and analgesia.
L 53 – Change "gen-eral" to "general".
Done
L 54 – I recommend replacing “dorsal decubitus" with "dorsal recumbency" (in the entire manuscript).
Done
At the end of the Introduction, a hypothesis should be stated.
Done
Materials and Methods
L 87-89 – tiletamine/zolazepam and romifidine were administered in one syringe or separately?
Tiletamine/zolazepam and romifidine were administered in one syringe
L 90 – "After achieving anaesthesia" - what does it mean? please describe more precisely how the state of achieving anesthesia was assessed.
Done
L 90 – "swine were cannulated into the jugular vein" - were the pigs really cannulated into the jugular vein? This is not usual in normal practice.
Yes, they were actually cannulated in the jugular as the limbs were bound by gauze.
L 92 – Change "ml/kg/h" to "ml/kg/hour".
Done
L 94 – "the volumes of each drug were increased to 40 ml" – In the LL/TT group, lidocaine/tramadol was diluted with saline to 40 milliliters? how many milliliters was given for the incisional and how many for the intraperitoneal block?
I specified in the text
Was the size of the hernia recorded? Were all the hernias the same size? Administration of the identical volume of drugs may produce a different response in hernias of different sizes, which can significantly affect the results.
I specified in the text
L 111 – I recommend replacing “systolic, diastolic, and mean blood pressure" with " systolic, diastolic, and mean arterial pressure" (in the entire manuscript).
DONE
L 112 – for "hemoglobin saturation" add "of oxygen".
DONE
L 111-112 – describe more precisely how HR, SAP, DAP, MAP and SpO2 was measured.
Done
L 115-116 – specify and add what T10, T15, T20, T25 and T30 mean. Are these times after dorsal recumbency?
They correspond to precise surgical moments to have the same surgical stimulus
L 118 – “Intraoperative surgical stimulus” – what does it mean? Please, describe exactly how to invoke it. Was it a skin incision? Reaction to peritoneum manipulation etc.?
they were incisions, manipulations and sutures of the various surgical planes
L 124-125 – How was the lidocaine administered?
Done
L 126-131 – When was postoperative pain assessed? Please complete the times.
Done
L 127 – "by three observers” – all at the same time or did all three participate in the assessment of all animals together?
Done
L 132 – according to which parameter the sample size selection was calculated?
The inclusion criterion was the presence of 3-5 cm diameters umbilical hernia inserted in materials and methods
On L 156, you state the variation, but it is not stated how it was calculated.
DONE
Exclusion criteria are not defined.
DONE
Results
I recommend describing differences "within groups" and "between groups" (throughout Results).
DONE
I recommend replacing "p = 0.000" with "p < 0.001" (throughout Results).
we have stated the actual p values given by SPSS
L 160 – "side effects" – how were the side effects assessed?
Delirium, allergic reactions, hyperthermia, hypothermia.
DONE
Tables
The legend to the tables should provide all the data so that the tables is comprehensible, and if it will be published separately from the main text.
DONE
What do the numbers mean? Mean/minimum/maximum? I recommend writing in the format mean (minimum – maximum).
DONE
HR in the LL group and SAP and DAP in the LT group are significantly higher in T1 (baseline). Why? I recommend commenting or explaining in the discussion.
Discussion
The manuscript is focused on the effects of lidocaine and tramadol. I therefore recommend focusing the discussion on these drugs, not on romifidine or zolazepam/tiletamine. I therefore recommend reducing or removing paragraphs on L 200-212, L 223-227 and L 231-240.
Done
Conclusions
The differences between LL, LT and TT described in Results are not mentioned here.
Done
The manuscript is beneficial for the professional public as well as for practice. In view of the above comments, however, I do not recommend accepting it in the present form. I recommend correcting and supplementing the manuscript and sending it for repeated review.
thank the authors for their careful proofreading and answers to my comments. Unfortunately, I still can't find or understand the following comments:
Keywords
L 37 – the words in the title and keywords are repeated.
Done
Introduction
At the end of the Introduction, I still do not find a stated hypothesis, what result the authors expect from the presented study.
Done
L 62 + entire manuscript – I recommend unifying liters and milliliters - "l, ml" or "L, mL" (according to the journal's requirements)
Done
L 81 – replace "magrophagic" with "macrophagic"
Done
Materials and Methods
L 153, 156 – I recommend unifying information on suture materials according to the journals requirements (formula, trade name, manufacturer, city, state)
Done
L 162 – the HR measurement method is not specified, please add it
Done
L 168-169 – T10, T15, T20, T25, T30 are times after dorsal recumbency? Please add to each item.
it's not the time after dorsal recumbency
Results
I recommend describing differences "within groups" and "between groups" (throughout Results and Tables).
If p = 0.000 and lower, I recommend writing p < 0.001 (in the entire manuscript).
Done
Tables
What does the data mean? Mean ± SD? Median/minimum/maximum? Please indicate the values for each table.
Done
HR in the LL group and SAP and DAP in the LT group are significantly higher in T1 (baseline). Why? I recommend commenting or explaining in the discussion.
Done
The manuscript is beneficial for the professional public as well as for practice. In view of the above comments, I recommend accepting it after minor corrections.
